# New Feature Selection Approach for Photovoltaïc Power Forecasting Using KCDE

**Jérémy Macaire [1,2,*], Sara Zermani [1,2] and Laurent Linguet [1,2]**

1   Espace pour le Développement (Espace-Dev), Université de Guyane, 97300 Cayenne, France; sara.zermani@univ-guyane.fr (S.Z.); laurent.linguet@univ-guyane.fr (L.L.)
2   Université de Guyane, DFR Sciences et Technologies, 97300 Cayenne, France
*   Correspondence: jeremy.macaire@univ-guyane.fr

**Abstract:** Feature selection helps improve the accuracy and computational time of solar forecasting. However, FS is often passed by or conducted with methods that do not suit the solar forecasting issue, such as filter or linear methods. In this study, we propose a wrapper method termed Sequential Forward Selection (SFS), with a Kernel Conditional Density Estimator (KCDE) named SFS-KCDE, as FS to forecast day-ahead regional PV power production in French Guiana. This method was compared to three other FS methods used in earlier studies: the Pearson correlation method, the RReliefF (RRF) method, and SFS using a linear regression. It has been shown that SFS-KCDE outperforms other FS methods, particularly for overcast sky conditions. Moreover, Wrapper methods show better forecasting performance than filter methods and should be used.

**Keywords:** photovoltaic power forecasting; feature selection; machine learning; Kernel conditional density estimator

## 1. Introduction

Energetic transition increases solar photovoltaic (PV) power integration in the energy mix as society monitors larger volumes of renewable energy [1]. Solar energy resource optimization is essential as it has huge potential. The International Energy Agency (IEA), in its Renewables 2019 report [2], estimates that in the next five years, solar PV systems will show the fastest growth among other renewables in the electricity sector [3,4]. However, this growth represents a severe threat to the grid operators because PV power is an intermittent resource [5]. PV power generation can drop or increase by more than 75% within 1 h [6]. Firstly, the intermittency of the resource makes it challenging to ensure the supply and demand balance [7], which can cause some problems concerning grid security [8]. Secondly, due to this intermittence, more maintenance operations are needed on the grid [9] as many plants must be regularly shut down or started up to ensure the supply and demand balance. This increases the failure rate, results in more extended maintenance, and the increased consumption of spares and replacement components [10]. All of this causes an increase in the PV power production costs [11] and carbon footprint [12].

These issues are significant in the ITCZ (Inter-Tropical Convergence Zone) because the climate can change quickly in these areas. In the ITCZ, solar irradiance variations can reach 800 W/m$^2$ within an hour [13].

As a solution, regional day-ahead PV power forecasting is proposed. It allows operators to manage energy resources more efficiently, decreasing the need for carbon-intensive reserves [14] and finally easing the integration of solar resources into the mix, reducing the costs associated with PV systems.

However, solar power forecasting research is immature [15,16]. This immaturity is also highlighted by the fact that some essential points of solar power forecasting are often passed over, such as feature selection (FS) [17].

FS is the process of selecting the best set of predictors [18–20], i.e., the features of the predictive model. The first objective of FS is to select the predictors that bring relevant information to improve the accuracy of the forecasts [21]. Too many features could lead to overfitting, and as some of these parameters are strongly correlated, they do not all bring relevant information. However, insufficient parameters could lead to underfitting due to the need for more information. Hence, an efficient forecasting method needs the most information with the least parameters. The second objective is to reduce computing time by removing redundant information. The feature selection aspect is significant because, with the same training data, an individual regression algorithm could perform better with different feature subsets [22]. We mainly focus on the FS methods used in solar forecasting, for which FS is a significant problem for model performance [23].

Day-ahead PV power forecasts depend on many meteorological parameters [24], enhancing the need for FS methods. This dependence is particularly true for the regional PV power forecasting model because meteorological conditions over several locations need to be considered.

In the earlier studies, only a few authors approached the FS [17], and most of these studies are recent. Despite a slight trend toward FS, it remains unused in most solar power forecasting studies, particularly for regional solar forecasts.

Most authors use statistical dependency calculation methods, termed filter methods, as FS. In [25], the authors used Pearson correlation to determine the most essential parameters to forecast the PV power production of a solar plant in South Korea. The problem is that Pearson correlation considers only linear correlation and cannot be used with more complex models. In [17], an RReliefF method is used as FS to predict short-term solar generation in the Netherlands. The RRF method allows the computing of weights to predictors, thus penalizing predictors that give different values to neighbors with the same response values and rewarding predictors that give different values to neighbors with different response values [26]. However, these statistical dependency calculation methods present some limitations, i.e., they do not consider the correlation between the predictors. Hence, a parameter with a low correlation but additional information could be discarded, and a parameter with a high correlation but with redundant information could be retained.

Another category of FS methods, termed wrapper methods, is sometimes used. The principle is to evaluate the impact of each parameter on the forecasting of PV power. Sequential Forward Selection (SFS) and Sequential Backward Selection (SBS) are the most known wrapper methods. Like the SBS [27], the SFS algorithm is a heuristic method that searches for the best predictors to minimize prediction error. The SFS starts with an empty set of predictors and, at each iteration, extends the previous set with the predictor, whose insertion gives the lowest prediction error [3]. However, even if wrapper methods seem very efficient, they have a high time complexity as they repeat the prediction process several times with different features.

In [3], an SFS is used with a linear regression (LR) to select the best set of parameters to make short-term predictions of Global Horizontal Irradiance (GHI) at the University Campus in Torino. However, LR, like the Pearson correlation, considers only linear correlation, and it is known that PV power forecasting is a non-linear problem. Some authors still use it because linear methods are often simpler to implement and provide better computational time [28].

To propose a method that overcomes the defects of earlier methods, which are linear correlation, high correlation with redundant information, and time complexity, we became interested in the combination of SFS and the Kernel Conditional Density Estimation method (SFS-KCDE).

Kernel Conditional Density Estimation (KCDE) is a probabilistic method often used in regression problems. KCDE could be helpful to FS due to its robustness, i.e., adding or removing a parameter does not drastically change the model, allowing for studying each parameter's impact on solar power forecasting. It is also a method adapted for non-linear problems. Furthermore, KCDE is very simple to implement and fully reproducible. There-

fore, KCDE presents all the essential characteristics of an efficient solar power forecasting model [29].

This study aims to propose a new FS method, using an SFS with a KCDE, to select the most relevant parameters from the Numerical Weather Prediction (NWP), which provides the most accurate PV power forecasts. The objective is to show that KCDE is a potential alternative for FS in the non-linear PV power forecasting problem. In this study, we will work on a regional day-ahead PV power forecasting model in French Guiana in the ITCZ.

A Physical and Artificial Neural Network (PHANN) model was used as a predictor with the selected parameters to validate the FS approach. PHANN is a hybrid method that combines a Numerical Weather Prediction model, which forecasts meteorological parameters, with a Neural Network, which uses the forecasted parameters to predict PV production. This FS method was compared to three others taken from the earlier studies. A Principal Component Analysis (PCA) has been added before the ANN to reduce computational time and to remove data noise.

First, we will discuss the data used in this study, and then the methods employed will be presented. The results and discussion will follow this.

## 2. Data

Here, we present the data used in this study. Historical PV power production data is used to implement our FS methods. Meteorological parameters, which are the features of our model, were day-ahead forecasted with a local Numerical Weather Prediction model. The meteorological model used is the Weather Research and Forecasting model (WRF) described in [30].

### 2.1. Historical PV Power Production Data

This study was carried out in French Guiana. All the data presented in this section were collected on the Open Data EDF website [31]. In French Guiana, there are 66 PV plants without storage, with an installed capacity of 46.46 MWc. PV electricity generation is highly centralized. Only eight cities have a PV power generation above 1 MW, which are Cayenne, Matoury, Rémire-Montjoly, Kourou, Macouria, Montsinéry, Sinnamary, and Saint-Laurent (Figure 1). These eight cities were the focus of this study.

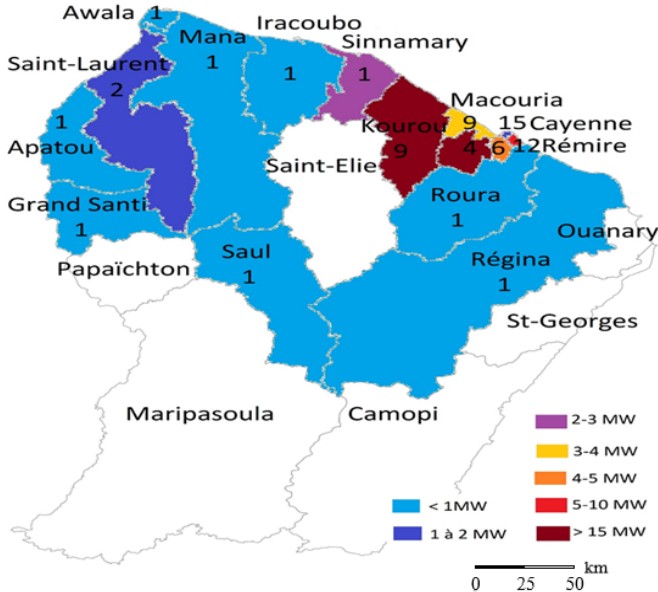

**Figure 1.** Map displaying the PV power production in French Guiana.

We collected the PV power generation (in MW) without storage from 2016 to 2020. These data present a timestep of 1 h.

*2.2. WRF Outputs*

PV power generation mostly depends on meteorological parameters. Hence, a Numerical Weather Prediction (NWP) model was used to forecast those meteorological parameters. This model was chosen because it is an open access model with a huge community (over 30,000 users registered). The forecasts of a global model were needed to be used as boundary conditions for the NWP local model. The Global Forecasting System (GFS) model was used as the input. GFS has the advantage of being open access. The data for 2016 were collected from the University Corporation for Atmospheric Research (UCAR) site [32].

The physical parameterization of WRF was taken from Diallo et al. 2018 [30] who adapted the WRF model to be more accurate for solar forecasting in French Guiana. Only the simulation parameters (timestep and spatial resolution) were changed to be able to run the model on local servers within a reasonable time limit.

We extracted 10 physical parameters from WRF (presented in Table 1) over 8 cities; hence, there was a total of 80 parameters.

**Table 1.** Meteorological parameters extracted from WRF.

| Parameter | Abbreviation | Unity |
|---|---|---|
| Global Horizontal Irradiance | GHI | $Wh/m^2$ |
| Global Horizontal Irradiance at the Top of the Atmosphere | GTOA | $Wh/m^2$ |
| Clear Sky Global Horizontal Irradiance | Gc | $Wh/m^2$ |
| Temperature | T | K |
| Wind speed towards east | U | m/s |
| Wind speed towards north | V | m/s |
| Surface pressure | PSCF | Pa |
| Humidity | Q | |
| Clear sky index | Kc | None |
| Clarity index | Kt | None |

The 10 physical parameters were: Global Horizontal Irradiance (GHI), Global Horizontal Irradiance at the Top of the Atmosphere (GTOA), Clear Sky Global Horizontal Irradiance (Gc), Temperature (T), the two components of horizontal wind speed U (positive for wind towards east) and V (positive for wind towards north), surface pressure (PSCF), Humidity (Q), the clear sky index (Kc), which is the ratio between GHI and Gc, and the clarity index (Kt), which is the ratio between GHI and GTOA.

## 3. Methods

Figure 2 shows a schematic representation of the proposed methodology. Firstly, meteorological parameters are extracted from WRF. Then, FS is applied to the extracted set of parameters to select the best features for day-ahead PV power forecasting. The FS method proposed is an SFS combined with a KCDE (SFS-KCDE) as the prediction model. This is wrapper method designed to consider correlations between features and to face linear problems. The details about SFS and KCDE are developed later in this section. Three methods have been used as a comparison because they are well known: the Pearson correlation, the RReliefF method, and an SFS with a linear regression. The hour has been added to the features dataset. Then, a PCA has been used on all sets of parameters selected to remove the features' noise and reduce computational time. PCA is a multivariate statistical method that extracts the critical information from the correlated features table and represents it as a new set of uncorrelated features [33]. To forecast regional day-ahead PV power, an ANN was trained with a different set of parameters chosen by each FS method to assess them and to compare their behavior on day-ahead PV power forecasting. ANN can recognize patterns in data and is now widespread in solar forecasting [34]. The ANN was set with 1 hidden layer with 10 neurons, with tangent sigmoid as the activation and Levenberg–Marquardt as the training functions. The historical PV power data were

used as targets to train the ANN to forecast regional PV power. In this case study, 243 days were available (February to September 2016). The model was trained on 203 days and tested on 40 days to keep the balance between having the most days possible for training and consistent testing. All the FS methods and prediction models were set with MATLAB. The RRF and the ANN were implemented using a toolbox. The other methods were set by implementing a code.

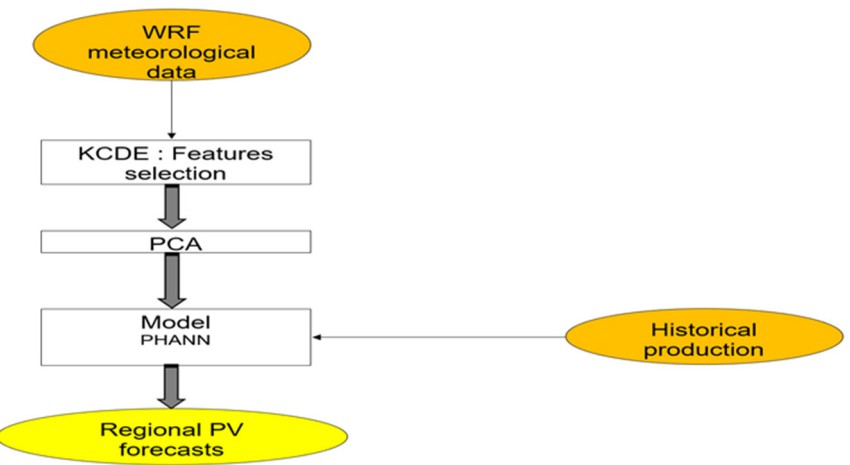

**Figure 2.** Schematic representation of the proposed methodology.

### 3.1. SFS

The SFS starts with an empty set of predictors and, at each iteration, extends the previous set with the predictor, whose insertion gives the lowest prediction error [3]. At each iteration, each parameter is individually added to the current feature set to form a new one, and PV power production is forecasted using a prediction model. Then, the parameter for which insertion gives the lowest rRMSE is retained. rRMSE was chosen because it is the most crucial metric for solar forecasting [32].

The algorithm of SFS is as follows in Figure 3.

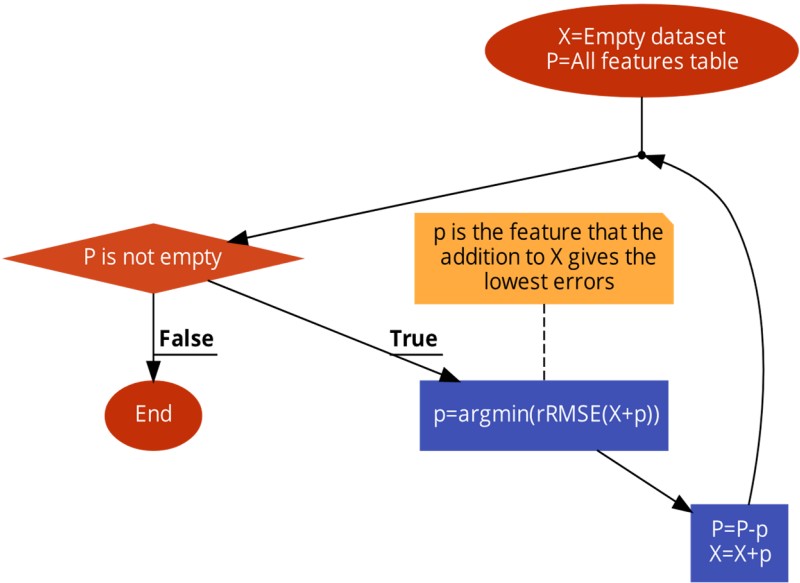

**Figure 3.** SFS algorithm.

Finally, a matrix X containing the selected features is obtained. Here, we propose the use of a KCDE as a prediction model inside the SFS, and this method is termed SFS-KCDE.

An SFS with a linear regression (SFS-LR) was also tested for comparison with the KCDE. The SFS was implemented using MATLAB.

### 3.2. KDCE

An SFS with a KDCE as a prediction model was used. KCDE [29] is a probabilistic method that allows for the prediction of unknown data by interpolating known data [35].

The principle is to estimate the Kernel density $K(h_x)$ for each parameter and then to provide a weight to each observation, as given below in Equation (1).

$$w_i(x) = \frac{K_{h_x}(||x - X_i||)}{\sum_{i=1}^{n} K_{h_x}(||x - X_i||)}$$

(1)

where $w_i$ is the weight of the $i$th observation; $x$ is the mean value of the parameter; $X_i$ is the value of the $i$th observation; and $n$ is the amount of data.

Then, the expectancy is calculated in Equation (2), which is the solar production according to the features, which gives the forecast.

$$\hat{m}(x) = \sum_{i=1}^{n} w_i(x) \times y_i$$

(2)

where $\hat{m}$ is the expectancy of the PV power production, and $y_i$ is the $i$th observation.

We used a Gaussian kernel for each parameter (Equation (3)).

$$K_h(||x - X_i||) = \frac{1}{h\sqrt{2\pi}} e^{-\frac{(x - X_i)^2}{2h^2}}$$

(3)

We calculated $h$ as follows (Equation (4)):

$$h = \left( \frac{4}{n \times (d + 2)} \right)^{\frac{1}{(d+4)}}$$

(4)

where $d$ is the total amount of features.

We chose this value for $h$ because it is the same used by Yang et al. 2019 [29].

Finally, as KCDE is more like a spatial exploration of the dataset, adding or removing a parameter will not drastically change the model, unlike the machine learning models, and hence, KCDE is a robust method. Furthermore, KCDE is a probabilistic method and considers non-linear correlation, unlike linear regression, and it is a fast and simple method.

### 3.3. Linear Regression

An SFS with a linear regression (LR), as in [3], was also tested to make a comparison. The LR is used to evaluate the different subsets. Linear regression is the simplest in computation as the tangent of the training data line is calculated and used when provided with new information. It does not have to update the model, which often results in faster training and forecasting compared to other machine learning models [36]. It estimates the linear relationship between the inputs and the target. The LR can be computed as follows in Equation (5) [37]:

$$y = \beta_0 + \beta_1 x_1 + \cdots + \beta_p x_p$$

(5)

where $y$ is the predicted PV power, $x_1, \ldots, x_p$ are the meteorological variables, and $\beta_0, \beta_1, \ldots, \beta_p$ are the linear parameters for each feature.

In this study, we will compare the behavior of an SFS with KCDE and an SFS with LR to see the impact of linear interpolation on FS.

### 3.4. Pearson Correlation

The Pearson correlation is a statistical index that provides a linear correlation between two parameters. It takes values between $-1$ and $1$, where 1 is a positive linear correlation, 0 is a lack of linear correlation, and $-1$ is a negative linear correlation. This method selected

the parameters which have the Pearson index furthest from 0. This is calculated as follows in Equation (6):

$$r_{xy} = \frac{\sum_{i=1}^{n}(x_i - \underline{x})\left(y_i - \underline{y}\right)}{\sqrt{\sum_{i=1}^{n}(x_i - \underline{x})^2}\sqrt{\sum_{i=1}^{n}\left(y_i - \underline{y}\right)^2}} \tag{6}$$

where the bar notation stands for the sample mean, $x$ is the feature, and $y$ is the output.

We selected features with $|r| > 0.1$ to remove only those parameters with very low correlation and select any parameter that could provide some information.

### 3.5. RReliefF

RReliefF is a filter method used to rank and select the best features. Three intermediate weights are defined given two nearest neighbors (Equations (7)–(9)). $W_{dy}$ is the weight comprising different values for the response $y$, which is the regional PV power production. $W_{dj}$ is the weight comprising different values for the same predictor, i.e., feature value, $F_j$. $W_{dy \wedge dj}$ is the weight of having different response values and different values for the predictor $F_j$. They are all set to 0 initially. Then, the RRF iteratively selects a random observation $x_r$, finds the k-nearest observations to $x_r$, and calculates for each nearest neighbor $x_q$ the new intermediate weights with the following equations:

$$W_{dy}^i = W_{dy}^{i-1} + \Delta_y\left(x_r, x_q\right) \cdot d_{rq} \tag{7}$$

$$W_{dj}^i = W_{dj}^{i-1} + \Delta_j\left(x_r, x_q\right) \cdot d_{rq} \tag{8}$$

$$W_{dy \wedge dj}^i = W_{dy \wedge dj}^{i-1} + \Delta_y\left(x_r, x_q\right) \cdot \Delta_j\left(x_r, x_q\right) \cdot d_{rq} \tag{9}$$

where $i$ is the iteration step number; $\Delta_y(x_r, x_q)$ is the difference in the value of the continuous response $y$ between observations $x_r$ and $x_q$; $\Delta_j(x_r, x_q)$ is the difference in the value of the predictor $F_j$ between observations $x_r$ and $x_q$; and $d_{rq}$ is a distance function.

Then, the final weight is calculated as follows (Equation (10)):

$$W_j = \frac{W_{dy \wedge dj}}{W_{dy}} - \frac{W_{dj} - W_{dy \wedge dj}}{m - W_{dy}} \tag{10}$$

where $m$ is the number of iterations by updates.

Finally, the weight $W_j$ represents the statistical impact of the feature on the PV power production. Positive weight implies that the feature and the output are statistically correlated. Negative weight implies poor correlation. Therefore, parameters with a $W_j > 0$ are selected.

This method was set as a comparison.

RRF was implemented using a MATLAB Toolbox.

## 4. Results and Discussion

We use three metrics to compare the accuracy of the models. The relative root mean square error (rRMSE), which penalizes great errors, the relative mean bias error (rMBE), and the relative mean absolute error (rMAE) are calculated as follows (Equations (11)–(13)):

$$\text{rRMSE} = \frac{\sqrt{\frac{1}{n}\sum_{i=1}^{n}(y_i - x_i)^2}}{\sum_{i=1}^{n} x_i} \times N \times 100 \tag{11}$$

$$\text{rMBE} = \frac{\sum_{i=1}^{n}(y_i - x_i)}{\sum_{i=1}^{n} x_i} \times 100 \tag{12}$$

$$\text{rMAE} = \frac{\sum_{i=1}^{n} |y_i - x_i|}{\sum_{i=1}^{n} x_i} \times 100 \tag{13}$$

where the $y_i$ are the solar irradiance forecasted values (W/m$^2$), the $x_i$ are the measured values of solar irradiance (W/m$^2$), and N is the amount of data.

### 4.1. Features Selected by Each FS Method

After collecting data from WRF, each FS method has been used on the dataset to select relevant features and discard the others. Figure 4 shows the selected features (in white) and the discarded features (in black) from the initial set by the different FS methods. Pearson correlation is the FS method that selected the most parameters. This is because only parameters with r > 0.1 are selected, which is a low value for r. Furthermore, 'V', which is the wind speed component (latitudinal speed) in French Guiana, has no linear correlation with PV power production. The humidity of only two cities has been selected by the Pearson correlation.

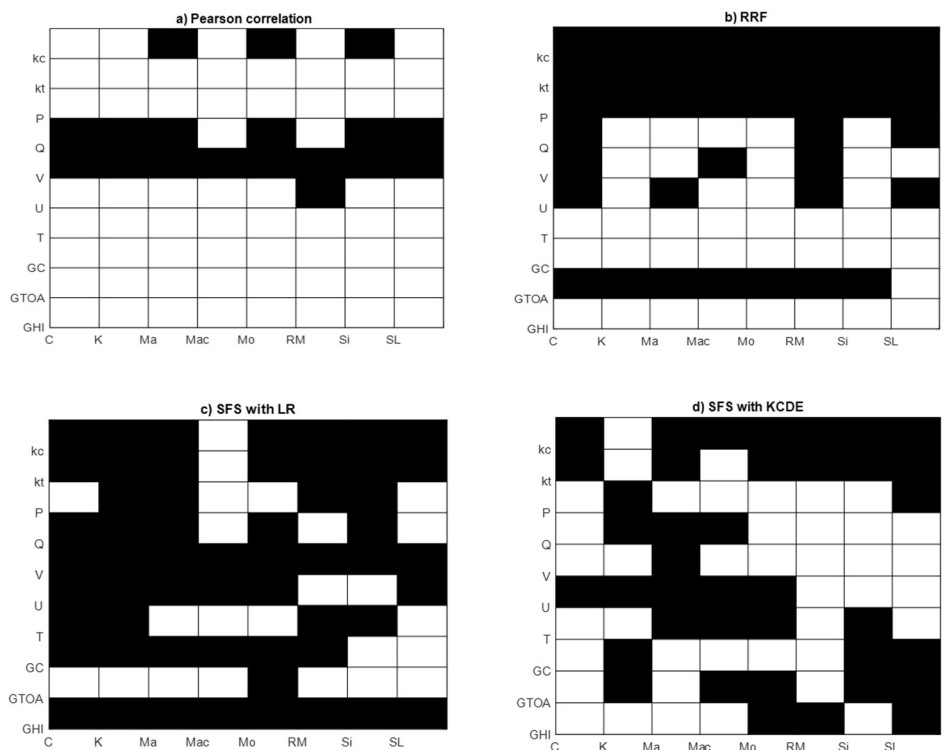

**Figure 4.** Plots depicting the parameters selected by (**a**) Pearson correlation; (**b**) RRF; (**c**) SFS with LR; (**d**) SFS with KCDE.

RRF mainly discarded surface pressure, kc, kt, and GTOA.

SFS with LR discarded all GHI and most of Gc, kc, and kt for the benefit of GTOA. These parameters are strongly correlated, and GTOA has a slightly better linear correlation with PV power production, particularly on sunny days, as it is not a forecasted but a calculated parameter. Additionally, GTOA is less spatially dependent than the other parameters, i.e., the GTOAs in different cities are more correlated than other parameters, which implies that GTOA is more consistent for regional forecasting. This explains the selection of GTOA over other irradiance parameters by the SFS-LS method. However, cloud motion and the variability of cloudy days will not be considered without GHI, kc, or kt.

*4.2. Comparison between Forecasting Errors for Each FS Method*

We analyzed the results for different sky conditions to observe potential behavioral differences between the FS methods.

Table 2 presents the metric results for the ANN trained with the features selected by each method and for overcast sky conditions (kc < 0.35). It is observed that for overcast hours, SFS-KCDE is especially efficient, with an rRMSE of 33.60% in comparison to between 36.75 and 37.93% for the others. This method also shows the rMBE closest to 0 and the lowest rMAE. Furthermore, for overcast hours, which need more complex models because of their variability, the SFS-LR shows the highest rMAE, which is undoubtedly due to the linear aspect of the method, unable to consider non-linear correlations. Effectively, as shown in Figure 3, SFS-LR discarded all the parameters containing information about clouds (GHI, kc, and kt), which is reflected by a poor prediction performance for overcast hours.

**Table 2.** Comparison between the different FS methods for overcast hours (kc < 0.35).

| Model | rRMSE% | rMBE% | rMAE% |
|---|---|---|---|
| ANN (without FS) | 37.02 | 6.47 | 27.27 |
| ANN (SFS-KCDE) | 33.60 | −0.62 | 25.61 |
| ANN (Pearson) | 37.01 | 10.42 | 26.43 |
| ANN (RRF) | 37.93 | 3.37 | 29.13 |
| ANN (SFS-LR) | 36.75 | 0.86 | 30.03 |

Table 3 presents the metric results for the ANN trained with the features selected by each method and for cloudy sky conditions (0.35 < kc < 0.65). It is observed that for cloudy hours, both the SFS methods are less accurate. Still, SFS-KCDE presents slightly better results than SFS-LR for the three metrics. Finally, for cloudy sky conditions, filter methods outperform wrapper methods.

**Table 3.** Comparison between the different FS methods for cloudy hours (0.35 < kc < 0.65).

| Model | rRMSE% | rMBE% | rMAE% |
|---|---|---|---|
| ANN (without FS) | 27.98 | −0.27 | 19.51 |
| ANN (SFS-KCDE) | 32.31 | 1.54 | 22.63 |
| ANN (Pearson) | 31.98 | −0.06 | 22.01 |
| ANN (RRF) | 29.09 | −0.90 | 20.00 |
| ANN (SFS-LR) | 32.69 | 1.94 | 22.82 |

Furthermore, for cloudy sky conditions, it is still observed that ANN without FS is the most efficient, with an rRMSE of 27.98% compared to 29.09 and 32.69% for the others. This method also shows the best rMAE, which means that for this type of sky condition, the balance between the amount of information and the number of parameters selected has yet to be reached by any FS methods. Adding more parameters should then be considered.

Table 4 presents the results of the metrics for the ANN trained with the features selected by each method and for clear sky conditions (kc > 0.65). It is observed that for clear sky hours, all the methods produce similar results, with a slight advantage for both of the SFS methods concerning rMAE, at around 19% compared to more than 20% for the other methods. It is also observed that for clear sky hours, SFS-KCDE shows the best rMBE with −3.30% compared to between −6.13 and −4.50% for the others. This method also shows the best rMAE with 19.08% compared to between 19.33 and 20.48%, which is certainly because, for clear sky hours, the correlation between the features and PV power production is less complex, and linear methods can be efficient enough. For rRMSE, only SFS-LR shows a slightly better number than SFS-KCDE. This is because, for clear sky conditions, linear correlation can be considered.

**Table 4.** Comparison between the different FS methods for clear sky hours (kc > 0.65).

| Model | rRMSE% | rMBE% | rMAE% |
|---|---|---|---|
| ANN (without FS) | 27.90 | −5.93 | 20.48 |
| ANN (SFS-KCDE) | 27.10 | −3.30 | 19.08 |
| ANN (Pearson) | 28.15 | −6.13 | 20.37 |
| ANN (RRF) | 27.29 | −5.16 | 20.22 |
| ANN (SFS-LR) | 27.06 | −4.50 | 19.33 |

Table 5 presents the metrics results for the ANN trained with the features selected by each method for all sky conditions.

**Table 5.** Comparison between the different methods for all sky conditions.

| Model | rRMSE% | rMBE% | rMAE% |
|---|---|---|---|
| ANN (without FS) | 28.34 | −4.87 | 20.57 |
| ANN (SFS-KCDE) | 27.96 | −2.64 | 19.70 |
| ANN (Pearson) | 28.95 | −4.90 | 20.75 |
| ANN (RRF) | 27.93 | −4.39 | 20.47 |
| ANN (SFS-LR) | 28.06 | −3.56 | 20.08 |

First, these results highlight the importance of FS. Except for the Pearson correlation, each FS method outperforms the prediction without FS, which is particularly true for regional forecasting, as there were initially 80 parameters, some of which were strongly correlated.

We observed that the Pearson correlation FS method performs poorly and is worse than the ANN without FS. This poor performance is because Pearson correlation only considers linear correlation between the parameters and removes features with non-linear correlations, which implies that the relation between the features and the PV power production is non-linear.

The RRF FS method has an rRMSE slightly better than that in the SFS with LR and the SFS with KCDE. However, the rMBE and the rMAE of RRF are higher than the rMBE and the rMAE of the two other methods because RRF does not consider the relations between the features. A feature with a low correlation output that brings complementary information will be removed.

The SFS-KCDE is slightly more efficient than the SFS-LR method. It particularly concerns the rMBE (−2.64% for SFS-KCDE compared to −3.56% for SFS-LR). The probabilistic nature of the KCDE could explain this.

Figure 5 shows the mean bias error for the SFS-KCDE according to kc and the cosinus of the solar zenith angle, cos(SZA). White cases mean no data in that range of values. It can be noticed that for values of cos(SZA) lower than 0.5, corresponding to the beginning and the end of the day, there is an alternation between overestimation (hot colors) and underestimation (cold colors). The importance of kc is reduced. For cos(SZA) values greater than 0.5, we observe an underestimation for the highest values of kc and an overestimation for the lowest values.

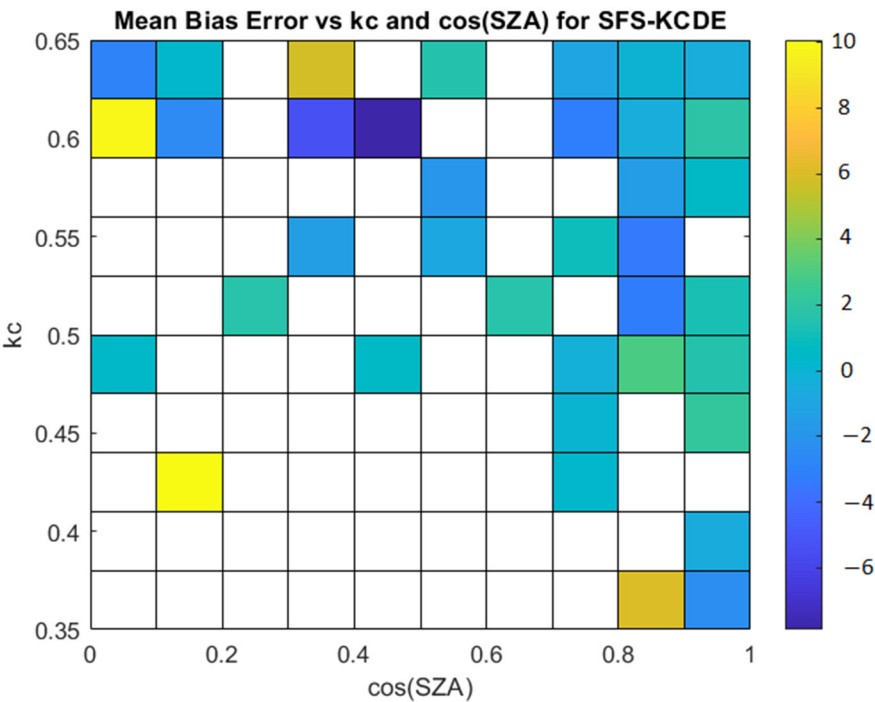

**Figure 5.** Plot depicting mean bias error vs. kc and cos(SZA) for SFS-KCDE.

### 5. Conclusions

In this study, we proposed a wrapper method termed SFS, combined with a KCDE to select the best meteorological parameters to forecast regional PV power in French Guiana. SFS was chosen to elude the problems of the filter FS methods. The SFS was based on KCDE to counter the non-linearity of PV power forecasting while maintaining the method's robustness, simplicity, and reproducibility. The SFS-KCDE method was compared to three other FS methods from earlier studies: Pearson correlation, RReliefF, and SFS with linear regression. The case without FS was also tested.

We showed that:

- SFS-KCDE is an adequate FS method as it improves forecast accuracy for clear and overcast sky conditions.
- Wrapper methods, like SFS-KCDE, show better forecasting performance than the filter methods and should be used.
- SFS-KCDE can face non-linear problems, as it outperformed other FS methods, including SFS-LR, for overcast hours.
- FS is an essential point of solar forecasting as it improves the forecast accuracy for most cases.

However, this SFS-KCDE method still has room for improvement. Firstly, simulations with more data should strengthen our study. Secondly, the method's benefits for operating PV plants should be tested.

Also, as the possible importance of cos(SZA) was highlighted, its influence should be investigated further.

Moreover, further studies on the criteria used in SFS to select features could ameliorate the accuracy of the forecasts. Here, only rRMSE was used as the criterion for SFS. A combination of different criteria for SFS will be studied in future work.

**Author Contributions:** Methodology, J.M.; software, J.M.; validation, J.M., S.Z. and L.L.; formal analysis, J.M.; writing—original draft preparation, J.M.; supervision, S.Z. and L.L. All authors have read and agreed to the published version of the manuscript.

**Funding:** This research received no external funding.

**Data Availability Statement:** Global Forecasting System are available at the University Corporation for Atmospheric Research at URL (accessed on 4 December 2019): https://rda.ucar.edu/datasets/ds084.1/. Historical photovoltaic power data are available at URL (accessed on 15 July 2020): https://opendata-guyane.edf.fr/explore/dataset/courbe-de-charge-de-la-production-delectricite-par-filiere/information/.

**Acknowledgments:** The authors are grateful to the FEDER-GUYANE for financing the SPESIS project.

**Conflicts of Interest:** The authors declare no conflict of interest.

## Nomenclature

| | |
|---|---|
| Abbreviations | |
| PV | Photovoltaïc |
| KCDE | Kernel Conditional Density Estimator |
| FS | Feature selection |
| SFS | Sequential Forward Selection |
| SBS | Sequential Backward Selection |
| RRF | RReliefF |
| WRF | Weather Research and Forecasting |
| ANN | Artificial Neural Network |
| PHANN | Physical and Artificial Neural Network |
| ITCZ | Inter Tropical Convergence Zone |
| NWP | Numerical Weather Prediction |
| PCA | Principal Component Analysis |
| LR | Linear regression |
| SFS-LR | SFS with linear regression as a prediction model |
| SFS-KCDE | SFS with KCDE as prediction model |
| GFS | Global Forecasting System |
| IFS | Integrated Forecast System |
| UCAR | University Corporation for Atmospheric Research |
| Symbols | |
| n | Amount of data |
| r | Pearson correlation index |
| rRMSE | Relative root mean squared error |
| rMBE | Relative mean bias error |
| rMAE | Relative mean absolute error |
| kc | Clear sky index |
| SZA | Solar zenith angle |
| Cos(SZA) | Cosinus of solar zenith angle |

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
