# Peer review of "New Feature Selection Approach for Photovoltaïc Power Forecasting Using KCDE"

_energies, doi:10.3390/en16196842_

Round 1

Reviewer 1 Report

The research is meanningful and useful. But some issues must be addressed before accepting for published in Energies.

(1) The abstract should be rewiritten and concised, especially to highlight the innovation of the researchï¼›

(2) The abbreviations tabel should be double-checked, some are not included in the table.

(3) In Line 141: The model WRF should add references;

(4) In Table1: Horizontal wind speed (U) means towards East and Horizontal wind speed (V) means towards North? If is true, please decrible clearly in the paper;

(5) Figure 5 should be detailed. It is best to refine the functions and parameters for each process node。 For examlpe: in the PCA, it is needed to describle the function;

(6) In Line 198-199: Why does the author choose 83 % of the data were used for training 198 and 17 % for validation? Please give detailed explanantion;

(7) In Line 213-218: I strongly suggest the authors give a flowchart for this algorithm, which will be more readable;

(8) In Line 268: and β0, β1,…, 268 βp the linear parameters for each features. The "β0" is omitted, please add it;

(9) In Line 408: "Tab. 5" correct "Table 5";

(10) Part "Discussion" should be rewritten. I strongly suggest authors combine this part with resulsts.

(11) In Line 452: “best meteorological parameters to forecast regional PV power in French Guiana.”, What's this sentence mean?

Reviewer 2 Report

English proofreading is mandated. 

Reviewer 3 Report

The document “Kernel Conditional Density Estimator in a Sequential Forward Selection as Feature Selection for day-ahead regional Photovoltaic power forecasting in French Guiana” study a proposed algorithm to forecast the PV production in French Guiana. The algorithm is based on Kernel Conditional Density Estimator which helps to obtain a small RMSE.

Comments

1.       The name of the manuscript requires modifications

2.       The abstract section needs to be improved. Some words are not well used.

3.       The introduction part is not well written. A lot of phrases have grammatical errors.

4.       The introduction is too large, only consider the most relevant information.

5.       In the methodology authors mention Matlab, but they do not give more information. They are using a toolbox, or implementing a code,

6.       The SFS was implemented in which software?

7.       The discussion part only mentions a comparison between the algorithms, but what about the benefits of the proposed algorithm in the efficiency of operating PV plants?

some sections require English revision. 

Round 2

Reviewer 1 Report

Thanks for author's revision. I think the manuscript can be accepted in present form.

Reviewer 2 Report

The authors have addressed my comments to my satisfaction.

English quality is acceptable.

Reviewer 3 Report

After reading the submitted manuscript entitled “New Feature Selection Approach for Photovoltaïc Power Forecasting using KCDE” it can be seem better structure and detailed information regarding the experimental part.

The new title of the manuscript has better fit and the abstract section now is reading easily. Furthermore, it can be mentioned that experimental and result sections were modified according to the reviewer comments. In this way, this work can be considered for publishing in the Energies Journal. I appreciate the authors works for addressing the suggested comments.